# A spatial memory signal shows that the parietal cortex has access to a craniotopic representation of space

**Mulugeta Semework[1,2,3]\*, Sara C Steenrod[1,2,3†], Michael E Goldberg[1,2,3,4,5,6,7]**

[1]Mahoney-Keck Center for Brain and Behavior Research, Department of Neuroscience, Columbia University College of Physicians and Surgeons, New York, United States; [2]Department of Neuroscience, Columbia University College of Physicians and Surgeons, New York, United States; [3]Division of Neurobiology and Behavior, New York State Psychiatric Institute, New York, United States; [4]Department of Neurology, Columbia University College of Physicians and Surgeons, New York, United States; [5]Department of Psychiatry, Columbia University College of Physicians and Surgeons, New York, United States; [6]Department of Ophthalmology, Columbia University College of Physicians and Surgeons, New York, United States; [7]Kavli Institute for Neuroscience, Columbia University, New York, United States

**\*For correspondence:**
mulugetas@gmail.com

**Present address:** [†]Institute of Clinical and Translational Sciences, Washington University School of Medicine, St Louis, United States

**Competing interests:** The authors declare that no competing interests exist.

**Abstract** Humans effortlessly establish a gist-like memory of their environment whenever they enter a new place, a memory that can guide action even in the absence of vision. Neurons in the lateral intraparietal area (LIP) of the monkey exhibit a form of this environmental memory. These neurons respond when a monkey makes a saccade that brings the spatial location of a stimulus that appeared on a number of prior trials, but not on the present trial, into their receptive fields (RFs). The stimulus need never have appeared in the neuron's RF. This memory response is usually weaker, with a longer latency than the neuron's visual response. We suggest that these results demonstrate that LIP has access to a supraretinal memory of space, which is activated when the spatial location of the vanished stimulus can be described by a retinotopic vector from the center of gaze to the remembered spatial location.
DOI: https://doi.org/10.7554/eLife.30762.001

## Introduction

Humans, and presumably monkeys, effortlessly establish a gist-like memory of their environment whenever they enter a new place. They can then use this memory to guide action even in the absence of vision. The hallmark of this environmental memory is that the objects remembered need not be relevant to the subject's current behavior. For example, although you may never be asked to point to the door with your eyes closed, you establish a memory of the door's location automatically. Although normally humans have no trouble pointing to objects in the room with their eyes closed, patients with bilateral parietal lesions cannot do this even though they can easily locate and point to objects in the room with their eyes open (**Levine et al., 1985**), suggesting an environmental memory impairment.

Visually responsive neurons in the frontal eye field (FEF) exhibit a signal that could represent environmental memory (**Umeno and Goldberg, 2001**). After monkeys make a number of saccades that bring a task-irrelevant probe stimulus into the receptive field of a visually responsive FEF neuron, many neurons respond on trials when a saccade brings the spatial location of the stimulus into the

receptive field, even though the probe stimulus did not appear on the current trial. However, because the investigators always used the same saccade both to establish and evaluate the memory response, it is not clear if the effect is a true spatial memory or merely a memory of receptive field stimulation associated with a saccade.

Here, we asked if neurons in the lateral intraparietal area (LIP), an area with visual, oculomotor, and mnemonic connections that serves as a priority map of the environment (*Bisley and Goldberg, 2010*), also exhibits an environmental memory signal. Indeed, we found that neurons in LIP did convey an environmental memory signal. Further, we found that the memory signal could be established even when the probe stimulus never appeared in the receptive field of the neuron, and occurred despite the fact that the monkey made a different saccade to bring the spatial location of the vanished probe stimulus into the receptive field. These results suggest that LIP has access to a representation of the visual world in at least supraretinal coordinates. Because saccades and reaching movements are coded in the parietal cortex in retinotopic coordinates (*Duhamel et al., 1992*; *Andersen et al., 1998*), these results suggest that one role of LIP is to transform a supraretinal representation of the remembered environment into a retinotopic representation more useful for the generation of action.

## Results

### Dataset

We recorded the activity of 131 neurons in three monkeys: 79 in Monkey A, 15 in Monkey B, and 37 in Monkey C. Because we had only a small number of cells in each monkey and the results from Monkeys B and C were comparable, these two data sets were combined for several analyses.

All neurons in our sample had visual responses to the onset of a saccade target in their receptive fields, and exhibited delay-period and/or presaccadic activity in a memory-guided delayed saccade task. Furthermore, we only studied the memory-related activity of neurons whose postsaccadic responses were not contaminated by the phasic and tonic components of an eye position signal

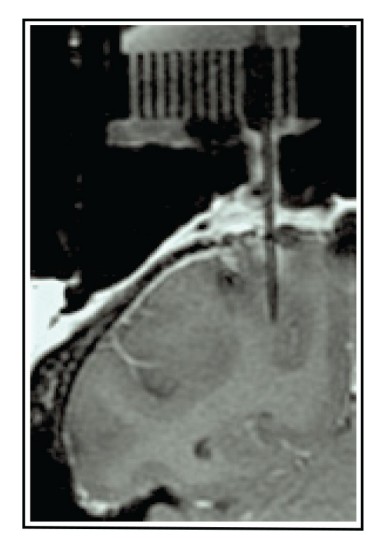

**Figure 1.** Location of LIP recordings. (**A**) A tungsten microelectrode (250 μm thick, straight shadow) located in the target area based on known LIP activity and the commonly used atlas-defined coordinates (*Paxinos et al., 1999*; *Saleem and Logothetis, 2012*) within the given coronal slice of the brain. This electrode location was at −2 AP and +10 ML.
DOI: https://doi.org/10.7554/eLife.30762.002

often found in parietal neurons (*Andersen et al., 1998*). All neurons were situated in the lateral bank of the intraparietal sulcus, as determined by structural MRI (*Figure 1*).

### Task 1: Basic memory task

We studied 72 LIP neurons using a task based on that used to demonstrate environmental memory in the frontal eye field (*Umeno and Goldberg, 2001*) (*Figure 2*). After determining the spatial tuning properties of the neuron being recorded, the monkey performed the basic memory task which is comprised of four blocks. We customized the arrangement of the stimuli (fixation point, saccade target, and probe stimulus) according to the spatial properties of each neuron's receptive field. In the first block, we asked the monkey to perform 20 visually guided saccade trials in which no probe stimuli appeared on the screen (*Figure 2*, Block 1). In these trials, no stimulus, including the saccade target, encroached on the receptive field of the neuron. Next, we asked the monkey to perform a block of 30 visually guided saccade trials in which a task-irrelevant probe stimulus appeared at a location that would be brought into the receptive field by the required saccade (i.e. the future receptive field of the neuron) (*Figure 2*,

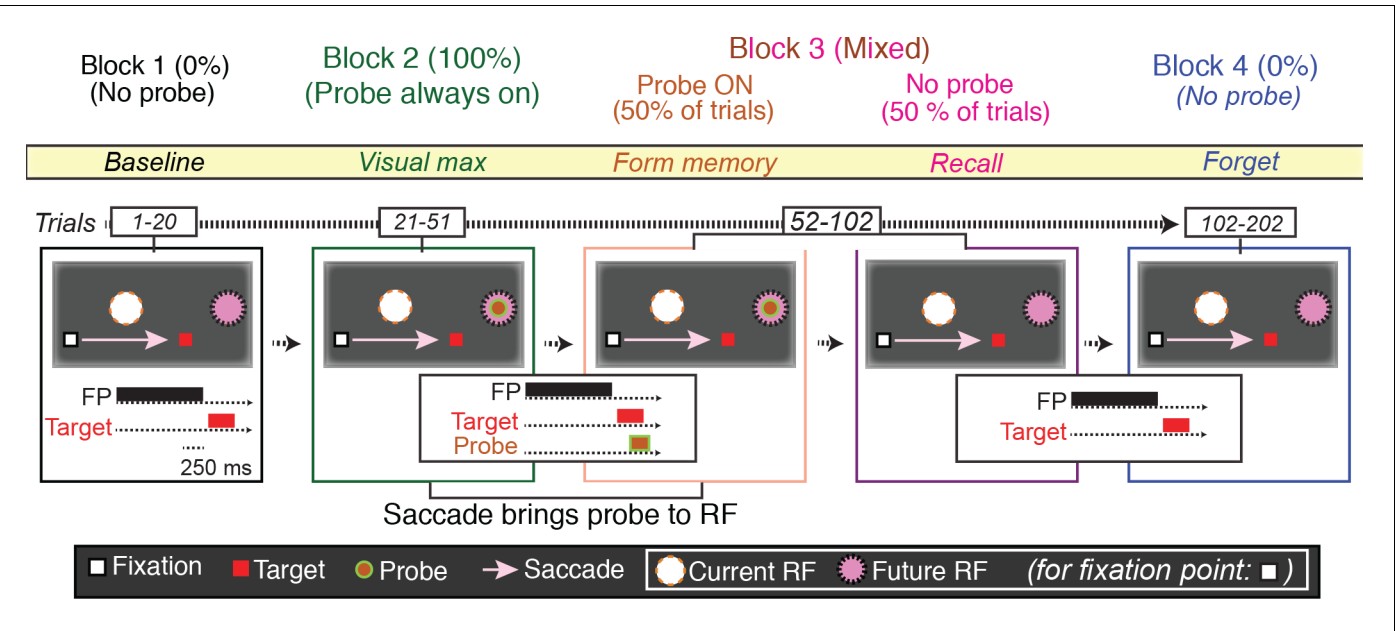

**Figure 2.** The basic memory task. The different trials types were performed in blocks. In Block 1 (baseline), the monkey made a visually guided saccade to a point outside the neuron's response field, to show that the saccade itself does not evoke neural activity. In the cartoon, the fixation point is a white square, the saccade target a red square, the presaccadic spatial location of the receptive field a white circle, the postsaccadic receptive field a magenta circle. The presentation times of the fixation point, saccade target and, when appropriate, probe stimuli are shown in the traces below. In Block 2 (visual max), the monkey made the same saccade, but now the saccade brought the probe stimulus into the receptive field. If the monkey made a saccade to the probe stimulus, the trial was terminated. In Block 3 (form and recall memory), for half of the trials, the probe stimulus appeared on the screen and was brought into the RF by the saccade. These trials were pseudorandomly intermixed with trials in which no probe stimulus appeared and the monkey made the same saccade. In block 4 (forget), the monkey made the same saccade but, as in Block 1, the probe stimulus never appeared.
DOI: https://doi.org/10.7554/eLife.30762.003

Block 2). Then, we pseudorandomly interleaved trials in which the probe stimulus appeared on the screen with those in which no probe stimulus appeared (*Figure 2*, Block 3). Finally, we presented the monkey with a block of 100 trials in which, again, the probe stimulus never appeared (*Figure 2*, Block 4).

22/49 of neurons in Monkey A (45%), and 11/23 (48%) exhibited environmental memory in the basic task: There was a significant difference ($p < 0.05$ by Wilcoxon rank sum) in the post saccadic response in Block 1 and the probe stimulus-absent trials in Block 3 (*Figure 3* for single cell and *Figure 4* for the population of all cells ($p<0.001$ Wilcoxon signed ran), black traces and raster symbols versus magenta traces and raster symbols). There was no significant post-saccadic response in Block 1 (*Figure 3*, for a single cell and *Figure 4* for the population of all cells black traces and raster symbols). There was a response to the probe stimulus in Block 2 when the saccade brought the probe stimulus into the receptive field (*Figure 3*, for a single cell and *Figure 4* for the population of all cells green traces and raster symbols). Neurons responded similarly in Block 3 as they did in Block 2 on the half of trials in which the probe stimulus appeared in the future receptive field (*Figure 3* for a single cell and *Figure 4* for the population of all cells, red traces and raster symbols). However, on the interleaved half of the trials in Block 3 in which a probe stimulus did not appear, the cells responded when the monkey made a saccade bringing the vanished probe location into the receptive field (*Figure 3* for a single cell and *Figure 4* for the population of memory cells, magenta traces and raster symbols). This response, which occurred after the saccade brought the spatial location of the probe stimulus into the receptive field even though no probe appeared on that trial, is the environmental memory response. Finally, after a block of forgetting trials in which the probe stimulus never appeared (Block 4 trials, identical to Block 1 trials for a single cell), the environmental memory response waned and, in some cases, disappeared entirely. The peristimulus time histogram (PSTH) figures do not include an average forgetting trace because the wide variation of forgetting traces made such an average uninformative

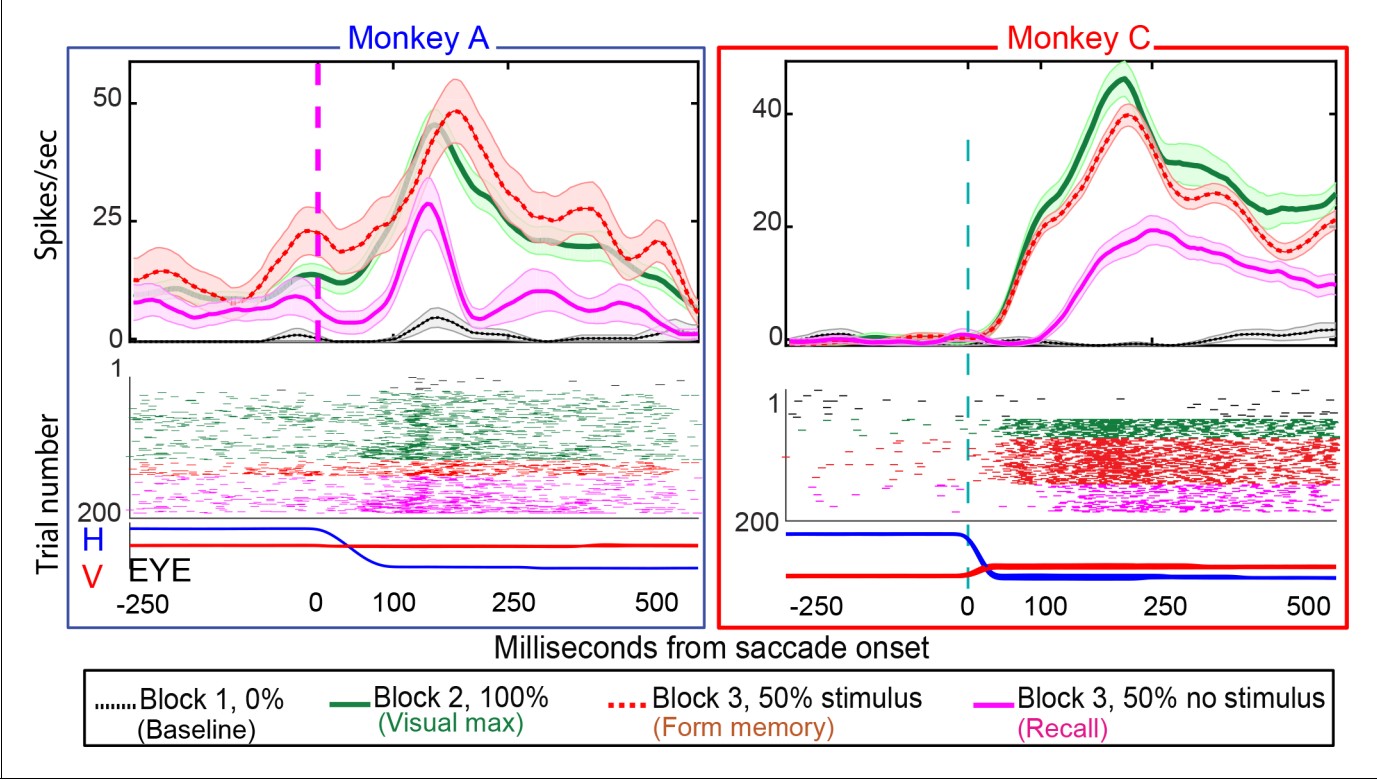

**Figure 3.** Single-cell responses in the basic memory task: from Monkey A (left panel) and Monkeys B and C (right panel). Neural activity aligned to the beginning of the saccade. Shaded regions in the peristimulus time histograms (PSTHs) are standard errors of means calculated using all the trials in the given block. Black traces and raster dots: activity in Block 1, stimulus absent. Green traces and raster dots: activity in Block 2, 100% stimulus present. Red (broken) traces and raster dots: activity in stimulus-present trials in Block 3. Magenta traces and raster dots: activity in probe stimulus-absent trials in Block 3. Here, there is a brisk response that shows the expected latency advance evoked by predictive remapping (*Duhamel et al., 1992*). Note that the latency of the memory response is nearly 100 ms longer after the saccade than the predictive response. The memory decays after many (up to 100) trials. H and V are horizontal and vertical eye movements smoothed using a 10 ms sliding causal filter.

DOI: https://doi.org/10.7554/eLife.30762.004

Both populations of neurons, including the statistically significant and the statistically non-significant neurons showed statistically significant memory responses (*Figure 5a*). The population showed a significant environmental memory response comparing the responses in Block one and the

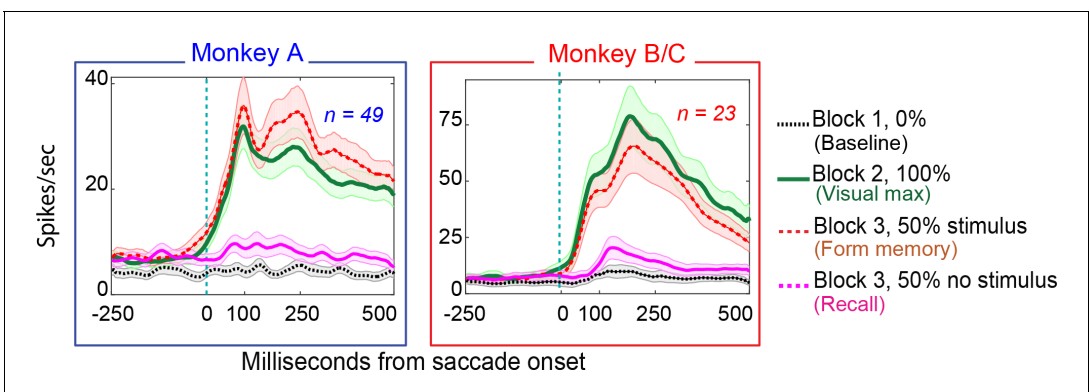

**Figure 4.** Population responses in the basic memory task: from Monkey A (left panel) and Monkeys B and C (right panel). Block descriptions are as in *Figure 3*.

DOI: https://doi.org/10.7554/eLife.30762.005

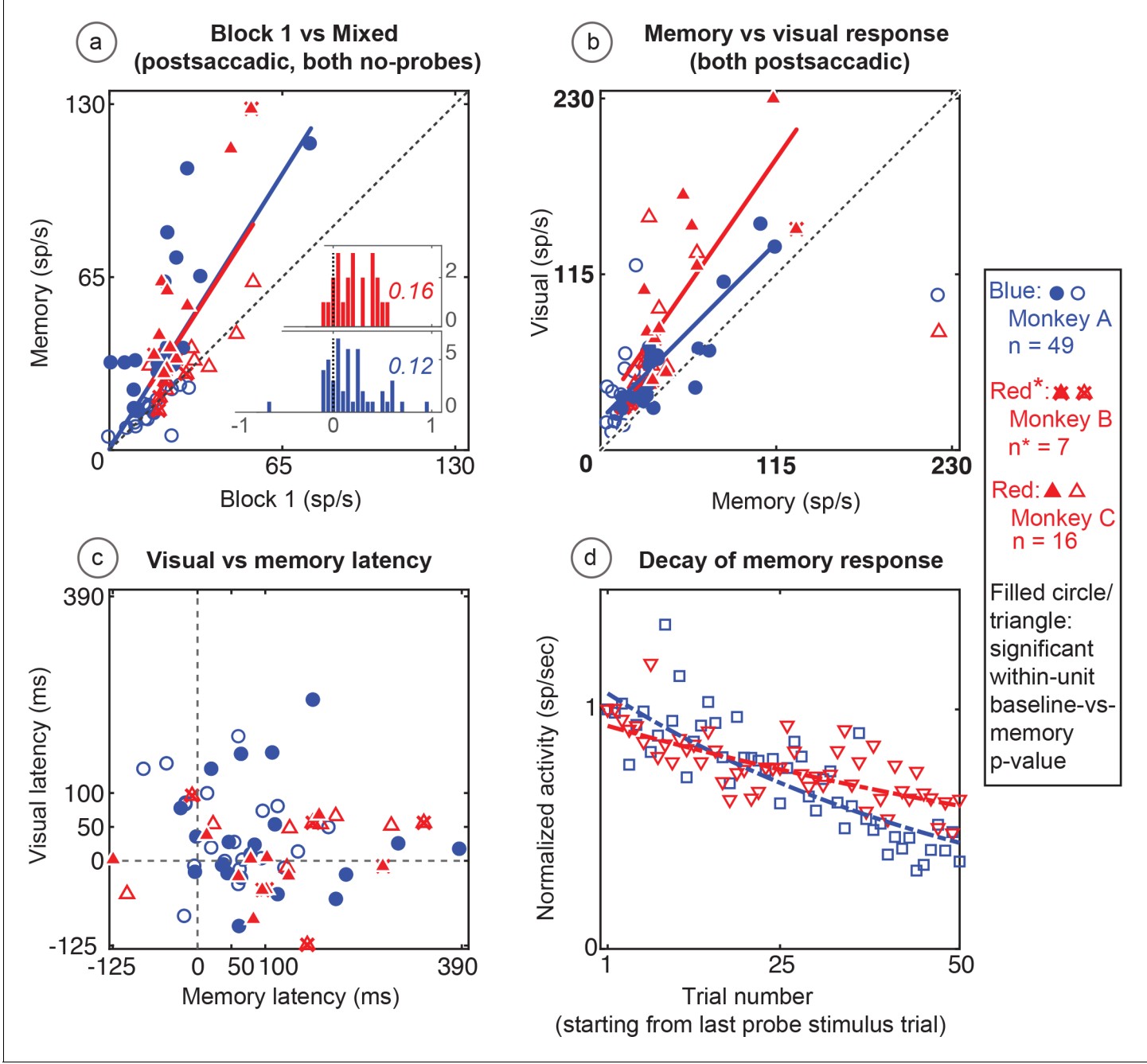

**Figure 5.** Population data for the basic memory task: For all recorded units (open circles and triangles) and for units which showed significant differences between baseline and memory blocks (filled circles and triangles, trial per trial peak comparisons, Wilcoxon signed rank, population p<0.001, within-cell Wilcoxon rank sum <0.05). **(A)** Comparison of peak activity in the baseline and memory conditions (Block 1 vs Block 3, no probe stimulus trials). Each point represents a single cell. Cells with memory activity lie above the x = y line (diagonal). Monkey A: blue circles; Monkeys B and C: red triangles (*=Monkeys B and C). Insets: index (Block 3 versus Block 1) histograms for the whole population, out of which the statistically significant units are shown in shaded circles and triangles on the scatter plot. A median (numbers above index histograms) and rank sum and KS test confirmed memory activity as they both showed that the index distribution for the parent population was positively skewed. **(B)** Peak memory activity plotted against visual activity. Blue and red lines in the scatter plots (**B and C**) are polynomial fits to each population data. They both indicate a positive shift of neural responses towards memory activity. **(C)** Latency of memory response versus the latency of visual response for cells. **(D)** Normalized decay activity. The memory response gradually decays after the last trial in which the probe appeared. The regression lines are data fits using a first order exponential. As indicated in 'D' legend, open squares and triangles are neural activities during decay trials compared to (normalized by) the mean of the last five trials with the probe stimulus (left y-axis).

DOI: https://doi.org/10.7554/eLife.30762.006

memory trials of Block 3 (Monkey A, p=0.0002 by Wilcoxon signed rank). Monkeys B and C (p=0.0003, Wilcoxon signed rank). (*Figure 5a*, Monkey A in blue and Monkeys B and C in red, significant cells filled symbols). The environmental memory response was weaker than the visual response evoked when the monkey made a saccade that brought the stimulus into the receptive field (*Figure 5b*, p<0.001 by Wilcoxon Signed rank). The latency of the environmental memory response was greater than the visual response evoked when a saccade brought the stimulus into the receptive field (*Figure 5c*) (Wilcoxon signed rank p=0.02 for Monkey A and B, and 0.0005 for Monkey C; mean visual/memory latency: 36/77 ms for Monkey A, 9/116 ms for Monkeys B and C). Of note some, but not all, of the cells exhibiting environmental memory had a shorter visual latency than one would expect given the minimal latency to a stimulus flashed in the receptive field. This latency advance is characteristic of the remapping response (*Duhamel et al., 1991*). However, because the memory responses for some neurons increased gradually, we were unable to establish memory latencies for all neurons using our latency method (for these neurons, we used other approaches, as described in the Materials and methods section).

During Block 4, in which the probe stimulus no longer appeared, the memory response gradually decayed (*Figure 5d*). On an individual cell basis, this decay (measured in the memory window as described in the Materials and methods section) was quite noisy and variable among cells. As a population, the normalized decay activity could be fit to a first-order exponential. To control for fast memory buildups and/or in-between block fluctuations, we normalized the decay activity with stimulus presentation block. Monkey A had a decay rate constant of 0.3 trials (i.e. versus stimulus/block-one trials, first-order exponential) ($R^2$ = 0.7). Monkeys B and C had similar results (decay rate constant = 0.13; $R^2$ = 0.6). For both monkeys and comparisons, the first and last 10 trials in the 50-trial forgetting block (Block 4) are statistically different (Wilcoxon signed rank, p<0001).

The memory cells were not just the expected tail of a population distribution. To investigate the population median shift, we used a classical median approach. We analyzed the distribution of index histograms and found that memory activity significantly increased the median (medians > 0.1 and compared with the index histograms with a mean-deducted version of the same distribution Wilcoxon signed rank p=3.7034e-12 for Monkey A and 1.4923e-06 for Monkeys B and C). We also used a non-parametric and robust skewness estimator called the medcouple (medc) measure (*Brys et al., 2004*). Unlike classical histogram descriptors such as skewness and kurtosis, this approach finds the scaled median difference between the left and right sides of a distribution. Its values range from −1 to 1 (left to right skewed, respectively). Analysis using this method for mixed trials in Block 3 (memory) versus Block 1 (baseline) indices (*Figure 5a* inserts) revealed that the population data was skewed toward a memory response (positive medc, ~1). The positive skewness of the memory responses (compared to baseline) is also indicated by an increased slope of a polynomial fit line (red and blue lines in *Figure 5a*) which is shifted to left of the identity line (black dashed line).

## Task 2: No-RF task

This task tested whether environmental memory could be evoked in the absence of receptive field stimulation. The basic memory task demonstrated that LIP neurons respond when a saccade brings the spatial location of a previously presented (now vanished) probe stimulus into their receptive field. However, because we established memory by having the monkey make a saccade that brought a stimulus into the receptive field, and then demonstrated the memory by having the monkey make the same saccade without a stimulus present, we could not know whether the memory response was independent of receptive field stimulation, or if it required visual stimulation or the receptive field to be established. Furthermore, we also could not determine whether the memory response was independent of the saccade used to establish it. To answer these questions, we used a modified version of the basic memory task, the No-RF task, in which the probe stimulus never appeared in the receptive field of the neuron and the monkey was required to make two different saccades on different trials (*Figure 6*).

Trials in the first block of the No-RF task were identical to those in Block 1 of the basic memory task. In Block 1, the monkey made a visually guided saccade (saccade 1) to a target outside of the neuron's receptive field and no probe stimulus appeared (*Figure 6*, Block 1). In Block 2, two trial types were pseudorandomly interleaved: no-probe trials identical to those in Block 1 (requiring saccade 1), and trials in which the probe stimulus appeared, but the monkey had to make a different saccade (saccade 2). In the latter trials, the probe stimulus appeared in the location corresponding

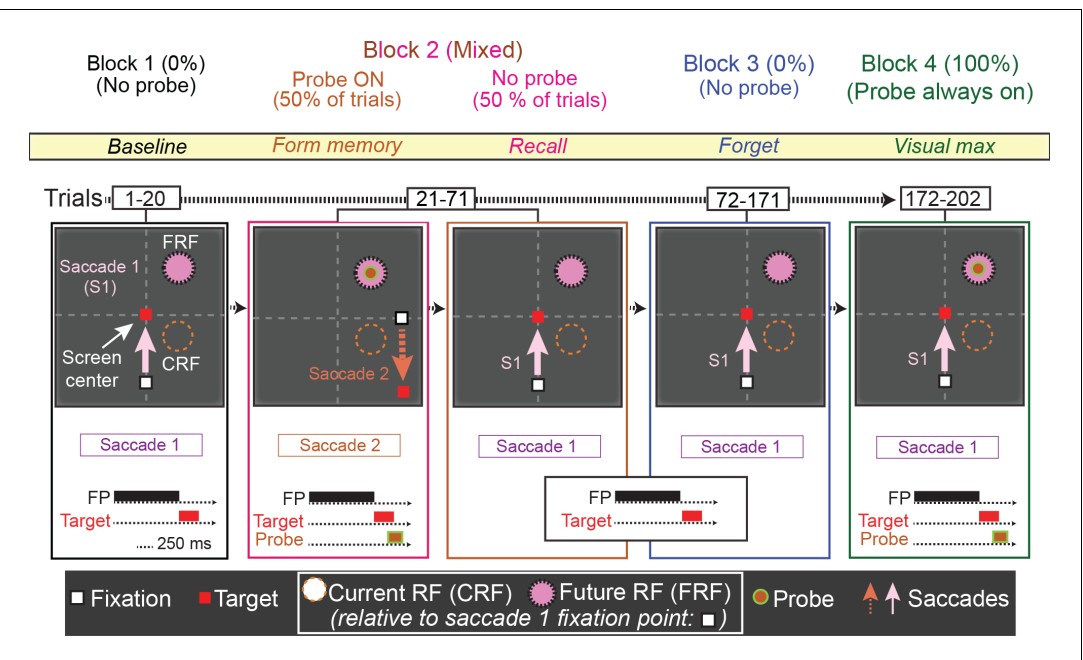

**Figure 6.** The No-RF task. Trials occurred in blocks. The first block (block 1, baseline) was used to establish the baseline activity of the neuron for saccade 1, prior to introduction to the probe stimulus. Block 2 (mixed, form and recall memory) introduced a task-irrelevant probe stimulus in half of the trials. In the half of trials where the probe stimulus appeared, the monkey was instructed to make a different saccade (saccade 2), which relocated the cell's receptive field away from the probe location (Block 2, orange). In the other half of trials, no probe stimulus was presented and the monkey made saccade 1, bringing the location of the (previously presented) probe stimulus into the cell's receptive field (Block 2, magenta). In Block 3 (forget, identical to Block 1), to measure the decay of the memory response, the monkey was instructed to make saccade one and no probe was presented. Finally, in Block 4 (visual max), to measure the visual response of the cell to the probe, the monkey was instructed to make saccade 1, and the probe stimulus was presented in the cell's receptive field.

DOI: https://doi.org/10.7554/eLife.30762.007

the neuron's receptive field after saccade 1. However, instead of making saccade 1, monkeys were instructed to make saccade 2, which relocated the neuron's receptive field away from the probe stimulus location, thus preventing it from ever visually stimulating the cell. Block 3 was identical to Block 1, allowing us to measure the decay rate of the memory response. In Block 4, the monkey was instructed to make saccade one and the probe stimulus was presented in the receptive field, visually stimulating the cell. This enabled us to compare the memory and visual responses (*Figure 6*, block 4 and *Figure 7*, green trace), and ensure that we had not lost the neuron during Block 3 while we observed the memory response decay.

We studied 44 neurons (30 in Monkey A, 14 in Monkey C) in the No-RF task, in which the stimulus never appeared in the cell's receptive field before and during the establishment of the memory. Some LIP neurons have a brief postsaccadic response to eye movements in all directions, including those away from their receptive fields (Wang and Goldberg, in preparation). Therefore, rather than excluding neurons with postsaccadic responses, we included neurons with small postsaccadic responses that did not differ between blocks 1 (saccades that will, in block 2, bring the spatial location of the vanished stimulus into the receptive field, but in which no stimulus ever appeared on the screen) and block 2 (different saccades that brought the receptive field away from the stimulus whose memory is being established, Wilcoxon signed rank: Monkey A p-value=0.4; Monkey C p-value=0.3). Although the stimulus never appeared in the neuron's receptive field, the cells responded when the saccade brought the spatial location of the vanished stimulus into the receptive field (*Figure 7*). The population exhibited a robust memory response (*Figure 8*, poststimulus histogram for all cells we studied). The population histogram shows, for one monkey, a small postsaccadic burst in the stimulus appearing trials of block 2. One could argue, by looking at the postsaccadic response in the trials where the monkey made the saccade that brought the visual stimulus to a location outside of our estimate of the cell's receptive field, that this postsaccadic response was a visual

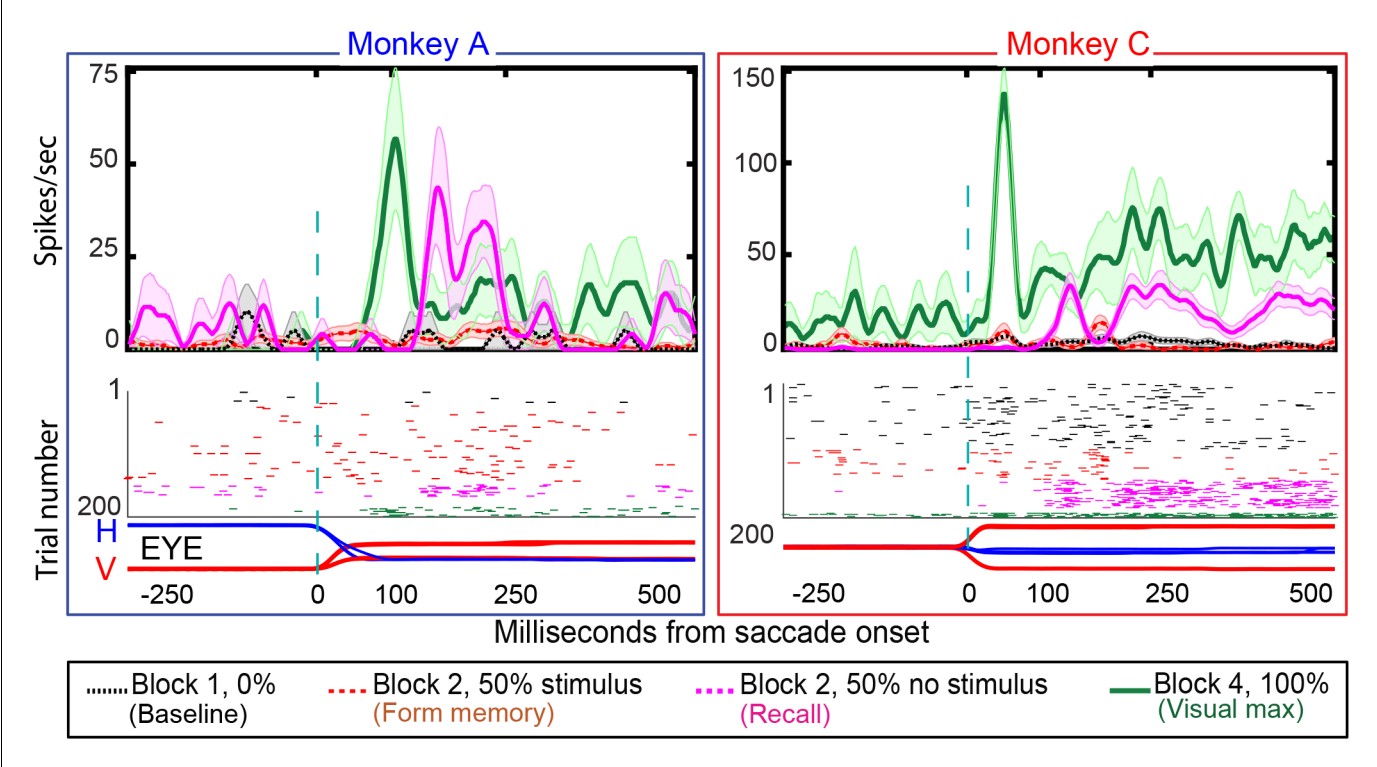

**Figure 7.** Single-cell responses in the No-RF memory task: from Monkey A (left panel) and Monkeys C (right panel). Block descriptions are as in *Figure 6*. Black traces and raster dots: activity in Block 1, stimulus absent. Red (broken) traces and raster dots: activity in stimulus-present trials in Block 2. Magenta traces and raster dots: activity in probe stimulus-absent trials in Block 2. The memory decays after many (up to 100) trials. Green traces and raster dots: activity in Block 4, 100% stimulus present. H and V are horizontal and vertical eye movements smoothed using a 10 ms sliding causal filter.
DOI: https://doi.org/10.7554/eLife.30762.008

response. However, we saw the same postsaccadic response in block 1, where the monkey made a different saccade and no visual stimulus had ever appeared in the receptive field. Thus, the identical postsaccadic response in block two cannot be a visual response.

26% (8/30) of Monkey A neurons and 57% of monkey C neurons (8/14) exhibited environmental memory in the No-RF task. (*Figure 7* for a single cell and *Figure 8* for the entire population of cells that we studied). The neurons showed a significant difference (p<0.05 by Wilcoxon rank sum) in the

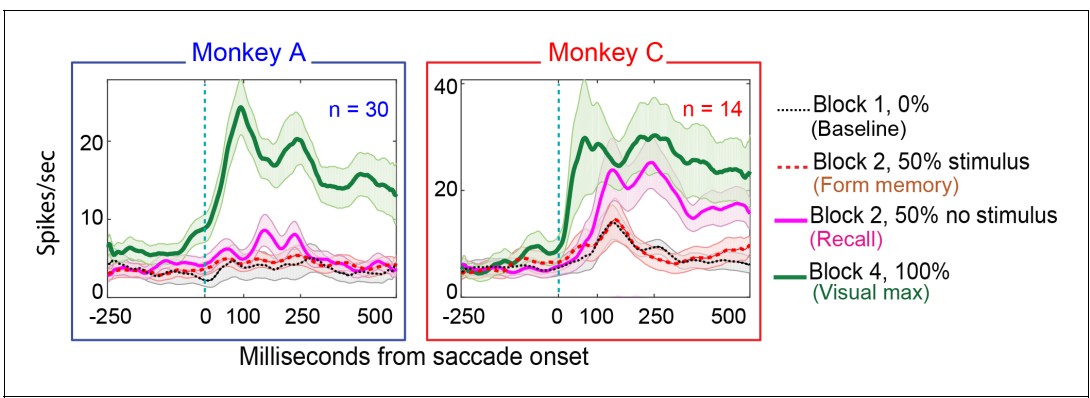

**Figure 8.** Population responses in the No-RF memory task: from Monkey A (left panel) and Monkeys B and C (right panel). Block descriptions are as in *Figure 7*.
DOI: https://doi.org/10.7554/eLife.30762.009

post saccadic response in Block 1 and the probe stimulus-absent trials in Block 2 (black traces and raster symbols versus magenta traces and raster symbols). They had a visual response in Block 4 when the saccade brought the stimulus into the receptive field. As in the basic task, this visual response was larger and had a longer shorter latency than the memory response. There was no significant difference in the postsaccadic responses in Block 1, when no stimulus appeared on the screen, and the memory forming trials in Block 2, when a stimulus appeared on the screen but did not appear in the receptive field of the neuron (*Figure 7*, for a single cell and *Figure 8* for the population of memory cells black traces and raster symbols) (Monkey A, p=0.4; Monkey C, p=0.3, Wilcoxon signed rank). Because the postsaccadic responses in Block 1 and the memory-forming trials in block to were not different, the much larger postsaccadic response in the memory trials of Block 2 could not have been a visual response.

Each population of neurons, including the statistically significant and the statistically non-significant neurons, taken as a whole, showed a statistically significant memory response (*Figure 9a*). The population showed a significant environmental memory response comparing the responses in Block 1 and the memory trials of Block 3 (Monkey A, p=0.006, Monkey B, p=0,007, Wilcoxon signed rank).

Similar to the basic memory task, the memory cells were not just the expected tail of a population distribution. The No-RF task population indices (mixed trials in Block 2 versus Block 1, *Figure 9a* inserts) were shifted to positive median values (median >= 0.1 and medc of 0.2 for both monkeys; Wilcoxon signed rank p=2.3087e-08 for Monkey A; and 0.007 for Monkey C). Here also, leftward shift (increased slope vs identity line) of a polynomial fit line (red and blue lines in *Figure 9a*) signified positive skewness of the memory responses (compared to baseline).

As in the basic task, the larger visual response in Block 4 was larger than the memory response in Block 2 (Monkey A, p<0.0001, Monkey C p=0.03 Wilcoxon signed rank). This shows that whatever decay we saw in Block 3 was not due to our losing the cell over this rather long experiment. The memory response had a longer latency than the neuron's visual response (*Figure 9c*) (Wilcoxon signed rank p=0.008 for Monkey A, and 0.006 for Monkey C; mean visual/memory latency: 39/101 ms for Monkey A, 25/106 ms for Monkey C).

As with data from the basic memory task, the normalized decay activity of the population (*Figure 9d*) could be fit to a first-order exponential. Monkey A had a decay rate constant of 0.09 trials and $R^2$ of −0.06; Monkey C, decay 0.07 trials, $R^2 = 0.12$. Significant p-values were found using a Wilcoxon signed rank test between trial mean comparisons of the first and the last 10 trials in the 50-trial decay epoch considered (Wilcoxon signed rank for Monkey A: p=0.0002, p=0.01 for Monkey C). The negative $R^2$ in Monkey A demonstrates the slowness of the decay and that a first-order exponential is not the best fit (it caused non-optimal line fit). This fit is used in the figures for consistency and visual clarity.

## Discussion

In this set of experiments, we demonstrate that LIP neurons exhibit an environmental memory signal, which lasts over a duration of multiple trials, decays slowly, and updates with eye movements. In addition, we demonstrate that a neuron does not require visual stimulation or a particular training saccade to establish memory activity. There is a similar signal in the frontal eye field (*Umeno and Goldberg, 2001*), in visual and visuomovement cells, but not movement cells. Those preliminary experiments, however, had two confounds. First, the memory was always established by stimulating the receptive field. Second, the saccade that evoked the memory was identical to the saccade used to establish it, so the memory could have been related to the saccadic process. In our experiments, we show that memory can be evoked by a stimulus that never appeared in the neuron's receptive field, and that the memory response manifested even when the saccade used to evoke it differed from the saccade used to establish it. Furthermore, as in the FEF (*Umeno and Goldberg, 2001*), memory activity occurred both in neurons which did exhibit saccadic remapping as well as those that did not. In sum, these results suggest that LIP has access to a supraretinal representation of the visual world despite the fact that all of its explicit responses are retinotopic, and that this memory response cannot be simply ascribed to saccadic remapping or saccade planning.

Our results can explain previous clinical observations regarding the role of parietal cortex in spatial memory. Patients with bilateral parietal lesions cannot point to an object in their room with their eyes closed, although they can easily do it with their eyes open (*Levine et al., 1985*). This suggests

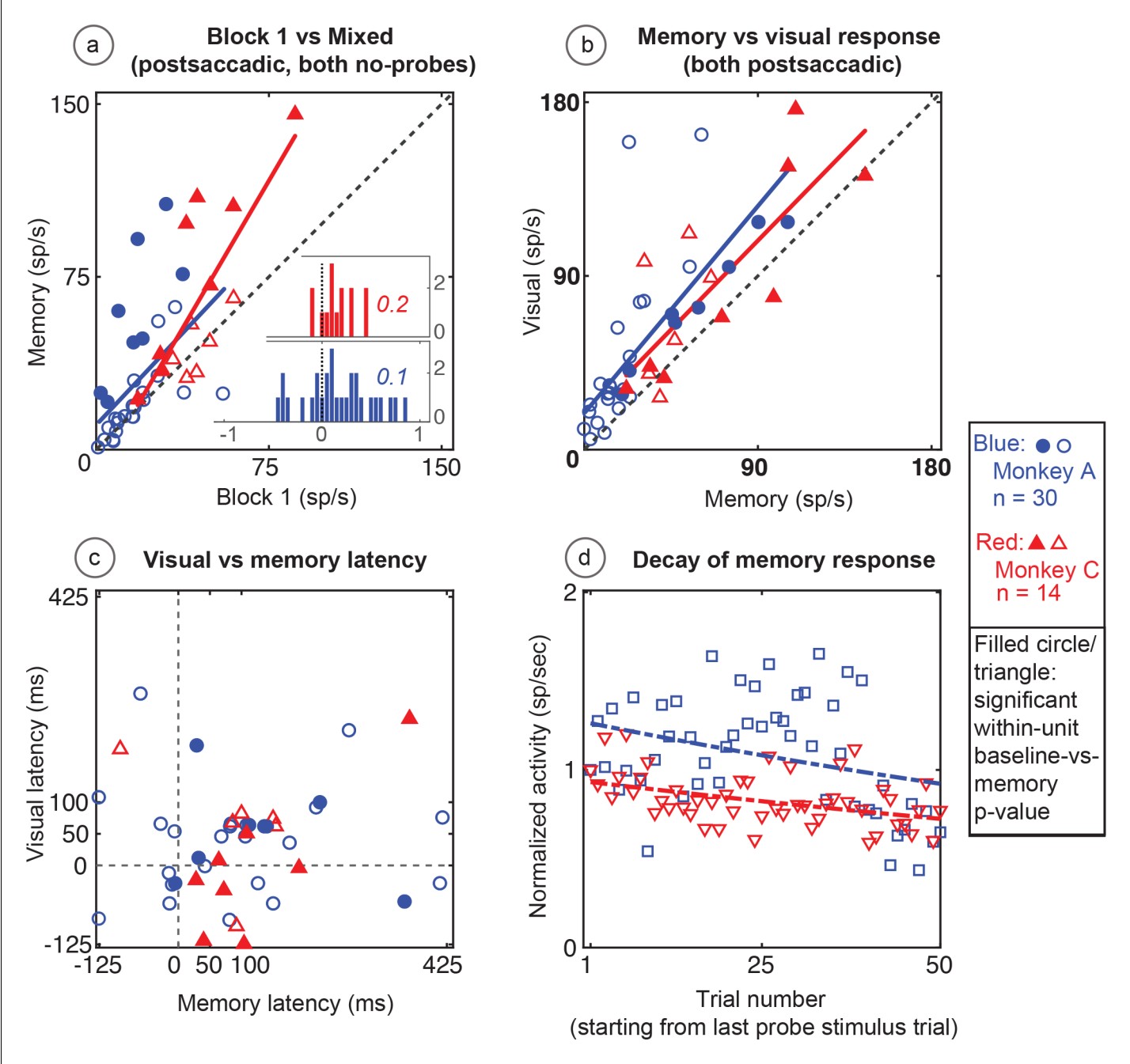

**Figure 9.** Population data in the No-RF memory task: All recorded units (open circles and triangles) and units which showed significant differences between baseline and memory blocks (filled symbols, trial per trial peak comparisons, Wilcoxon signed rank, population p<0.001, within-cell Wilcoxon rank sum <0.05). (**A**) Comparison of peak activity in the baseline and memory conditions (Block 1 vs Block 2, no probe stimulus trials). Each point represents a single cell. Cells with memory activity lie above the x = y line (diagonal). Monkey A: blue circles; Monkey C: red triangles. Insets: index (Block 2 versus Block 1) histograms for the whole population, out of which the statistically significant units are shown in shaded circles and triangles on the scatter plot. A median (numbers above index histograms) and rank sum and KS test confirmed memory activity as they both showed that the index distribution for the parent population was positively skewed. (**B**) Peak memory activity plotted against visual activity. Blue and red lines in the scatter plots are polynomial fits to each population data. They both indicate positive shift of neural responses towards memory activity. (**C**) Latency of memory response versus the latency of visual response for cells. (**D**) Normalized decay activity. Memory response gradually decays after the last trial in which the probe appeared. The regression lines are data fits using a first order exponential. As indicated in 'D' legend, open squares and triangles are neural activities during decay trials compared to (normalized by) the mean of the last five trials with the probe stimulus (left y-axis).
DOI: https://doi.org/10.7554/eLife.30762.010

that these patients do not have access to a remembered spatial representation of their environment. More importantly, patients with right parietal lesions report their spatial memory in a retinotopic frame. *Bisiach and Luzzatti (1978)* asked two Milanese patients with right parietal damage to describe their memory of the Piazza del Duomo in Milan. They found that these patients could only describe the recalled environmental landmarks that would fall in their unaffected (right) hemifield, which depended on their (imagined) vantage point. For example, when they described Piazza del Duomo as if they were standing with their back to the cathedral they remembered only the landmarks to the right of the cathedral. However, when they described the Piazza del Duomo as if they were facing the cathedral they only remembered the landmarks on the other side of the square, which were now to the right of their imagined vantage point. This experiment demonstrates two key ideas; (1) The brain stores long-term memory of environments in supraretinal coordinates in an area unaffected by a parietal lesion, since the patients could, under certain circumstances, remember the entire Piazza del Duomo. (2) Retrieving that memory requires the parietal cortex, but the right parietal patients could only report (and therefore mentally represent) objects on the right side of their imagined vantage point. They had a mnemonic left hemianopia for a remembered environment. If their memory were reportable in a supraretinal frame they should have been able to report the entire scene. Because they could not do so, and could only report objects in the non-neglected hemifield, the report of their environmental memory must be processed by the retinotopic map of the parietal cortex. Our results are consistent with this result. The memory of an environment can be stored a supraretinal frame, but must be reported by neurons with retinotopic receptive fields. The neurons that we have reported here fit that description. They have access to a supraretinal representation of the visual field, because environmental memory does not require stimulating their receptive fields, but they only report that memory when the spatial location of the remembered stimuli lies in their receptive field.

The memory response has both a longer latency and a lesser intensity than a simple reafferent visual response. This suggests that it does not arise from a feedforward mechanism, just as task-related pattern selectivity in V4 requires a late, presumably feedback, mechanism (*Ipata et al., 2012*). There are two known mechanisms by which LIP can create a spatially accurate representation of space despite a constantly moving eye. The first is through remapping, in which a neuron will respond immediately after a saccade brings the spatial location of a recently vanished stimulus into its receptive field (*Duhamel et al., 1991*; *Sun and Goldberg, 2016*). This effectively changes the spatial origin of the retinotopic coordinate system from the presaccadic fixation point to the saccade target. The next saccade could just as easily remap the current retinotopic representation into a coordinate system centered on the next saccade target, and so on, through a number of saccades. This is unlikely to be the basis of our results, however, because many cells that show environmental memory do not show pre-saccadic remapping when a saccade brings a stimulus into its receptive field.

The second mechanism creates a spatially accurate retinotopic representation from a supraretinal representation. There is excellent human psychophysical evidence that the saccadic system has access to a supraretinal representation of space that does not involve remapping. *Karn et al. (1997)* showed that there is little difference in the inaccuracy of memory-guided saccades after two or six intervening visually-guided saccades. Reliance on a remapping mechanism for accuracy would entail the concatenation of errors from saccade to saccade, whereas access to a supraretinal coordinate frame would enable a more stable representation of space across multiple saccades. In a similar experiment, *Poletti et al. (2013)* asked humans to make different numbers of intervening saccades between the presentation of a target for a memory-guided saccade and executing the actual saccade. For the first few intervening saccades, the variability of the memory-guided saccade increased linearly, as expected if the error of each remapping process would concatenate. However, after the first few saccades, the increase in variability decreased with each subsequent saccade, as if the representation of the saccade target had shifted to a supraretinal frame which would not increase its error with each saccade. To explain this phenomenon, they proposed a model that entails a shift from an early remapping mechanisms to a subsequent supraretinal mechanism. This late supraretinal representation could arise from a gain-field mechanism. Visual responses in LIP are linearly modulated by the position of the eye in the orbit, the gain field. The eye position signal that modulates the gain fields in LIP could come from the representation of eye position in area 3a (*Zhang et al., 2008*). It is mathematically trivial to calculate target position in supraretinal coordinates from

retinotopic gain fields (*Zipser and Andersen, 1988*; *Salinas and Abbott, 1997*; *Pouget and Sejnowski, 1994*). Although gain fields are inaccurate immediately after a saccade (*Xu et al., 2012*), they could contribute to a late, accurate, supraretinal representation that develops when the eye position signal is accurate, as one would expect if the eye position signal arose from proprioception. The neurons that we studied did not have eye position modulation of their visual responses – such neurons often have a phasic and tonic postsaccadic response that arises from the eye position signal itself (Wang et al.). Indeed, because the memory of the stimulus was established well before the saccade that evoked the memory, the supraretinal representation of the remembered stimulus could easily be implicitly encoded in the gain-field responses of LIP neurons (*Zipser and Andersen, 1988*).

Data from patients with parietal lesions, such as the previously described results from Bisiach and Luzzatti, suggest that although LIP might calculate target position in retinotopic coordinates, the parietal cortex does not store the results of that calculation. Instead, a supraretinal representation of space could be stored elsewhere, possibly in the medial temporal lobe. O'Keefe discovered that neurons in the rat hippocampus (the 'place cells') discharge when rats enter spatial location that they have seen before (*O'Keefe and Dostrovsky, 1971*). Other studies have duplicated this result in the monkey hippocampus (*Ono et al., 1993*; *Rolls et al., 1989*). Grid cells in entorhinal cortex are thought to be the precursors of hippocampal place cells (*Moser et al., 2008*) and these have been found in the monkey (*Buffalo, 2015*) as well as in humans (*Jacobs et al., 2013*). It is known that there is a direct, reciprocal connection between the parahippocampal gyrus and LIP (*Suzuki and Amaral, 1994*; *Baizer et al., 1991*). Thus, LIP could send an eye-position modulated visual signal to the medial temporal lobe, which could serve as a building block for a supraretinal memory. The medial temporal lobe could then use this activity to generate the supraretinal component of place cell activity and send this supraretinal signal back to LIP. LIP itself would only be activated when the spatial location of the vanished stimulus can be described by a retinotopic vector from the center of gaze to the receptive field of the neuron. Thus, LIP has access to a supraretinal representation, but expresses only a retinotopic representation. It is important to emphasize that LIP is part of a spatial network that includes the FEF and the superior colliculus. Although neuropsychological studies suggest that parietal damage affects spatial memory, we do not know if damage to other parts of the network has a similar effect.

Although early models posited that the oculomotor system uses a supraretinal representation (*Zee et al., 1976*), it is clear that the parietal cortex sends a spatially accurate retinotopic signal to the oculomotor (*Pare and Wurtz, 1997*) and skeletomotor (*Batista et al., 1999*) systems. Even auditory stimuli, which are initially coded in craniotopic coordinates, are converted to retinotopic coordinates for the oculomotor (*Jay and Sparks, 1987b*; *Jay and Sparks, 1987a*) and skeletal motor system (*Grunewald et al., 1999*; *Cohen and Andersen, 2000*) in parietal cortex, and the oculomotor system of the superior colliculus (*Jay and Sparks, 1984*). We suggest that the environmental memory signal in LIP is an example of the conversion of a supraretinal signal to a retinotopic signal for memory and spatial attention as well as for motor control.

## Materials and methods

The Animal Care and Use Committees at Columbia University and the New York State Psychiatric Institute approved all of the animal protocols in this study as complying with the guidelines established in the United States Public Health Service Guide for the Care and Use of Laboratory Animals (protocol NYSPI-1225-C). We used one female and two male rhesus monkeys (Macaca mulatta) with weights between 6 and 13 kg.

We first trained monkeys to sit in a primate chair using a pole and collar technique. After chair training, we surgically fitted the monkeys with acrylic implants, anchored with titanium or ceramic screws, outfitted with a headpost which stabilized the monkeys head while sitting in a primate chair. During the same surgery, we implanted scleral search coils around both eyes (*Judge et al., 1980*) and brought the coil wire out to a plug on the acrylic. After basic behavioral training, we performed a craniotomy to expose the intraparietal sulcus (using coordinates determined from a structural MRI), over which we affixed a recording chamber in the acrylic.

We located the actual recording sites in each monkey by performing MRIs with the recording electrode in place (see *Figure 1*). We tranquilized the monkey in the vivarium with ketamine and atropine for transport to the MRI lab. We then anesthetized the monkey using endotracheal

isoflurane, and positioned it in a Kopf MRI-compatible stereotaxic instrument. To create the images, we used a custom-made rigid and flexible RF array coils using GE MR750(3T) and GE Signa(1.5 T) scanners. We used custom Matlab software, Osirix, GE's MediaViewer, and DicomWorks to analyze the data.

We performed all surgeries using ketamine/isoflurane general anesthesia using aseptic surgical technique. The monkey recovered fully before testing restarted. During testing, monkeys worked for their daily water intake and were supplemented with dried and fresh fruits. We monitored monkeys' weights and general health on every recording day, and at least once a week.

We controlled all experiments using the REX (downloadable from: https://nei.nih.gov/intramural/software) system (*Hays et al., 1982*) and recorded single-unit activity with glass-insulated tungsten electrodes introduced through a guide tube positioned plastic grid with 1 mm spacing between possible penetrations (*Crist et al., 1988*). In general, recording sessions lasted between 5 and 8 hr depending on the stability of the cell and the monkey's willingness to continue to work. We took care to avoid making penetrations with the electrode in adjacent grid holes from the previous day's penetration in order to reduce tissue damage. We kept careful logs regarding the location and depth of each penetration, an estimate describing cell types encountered, and the quality of the monkey's behavior. During every testing and training session, we monitored monkeys using a closed-circuit camera and monitor.

We verified stimulus timing using a photoprobe affixed to the screen, which emitted a pulse for each event. We monitored the monkey's eye movements using a CNC phase detector (Crist Instruments, Hagerstown, MD) to decode the search coil signal (*Judge et al., 1980*) and sampled the signal at 1 kHz. During testing, monkeys sat in a primate chair in a sound-attenuated Faraday room. We performed these experiments in two experimental setups. In one set up, the monkeys sat 57 cm away from a CRT monitor (ViewSonic Professional Series P225F) with a refresh rate of 120 Hz. In the second setup, we presented stimuli with a Hitachi CPX275 LCD projector with a refresh rate of 60 Hz, on a screen 72 cm from the monkey.

Once we isolated an LIP neuron and confirmed that it displayed canonical visual, delay, and perisaccadic responses, we probed the boundaries of LIP neuron receptive fields by having monkeys make memory-guided saccades (*Hikosaka and Wurtz, 1983*) to targets appearing in locations which were inside and outside of the cell's receptive field. In some cases, we used a fixation task to characterize the cell's receptive field by having the monkey fixate a point while stimuli briefly flashed (50 ms) pseudo-randomly on the screen, generating a map of locations that evoked visual responses (measured 50–150 ms after the flash). We ensured that saccade target and probe stimuli locations were positioned correctly with respect to future and current receptive field locations to avoid unintentional contamination by a visual response.

We transformed the REX data into a form analyzable by MATLAB (MathWorks, Natick, MA) using the REXTOOLS software (downloadable from https://nei.nih.gov/intramural/software). Although the REX system provides an estimate of activity using the MEX (downloadable from https://nei.nih.gov/intramural/software) spike sorter, for the analyses in this paper we used the MEX system to digitize the neural data and sorted the spikes offline using the MEX offline analysis program. The custom codes, and raw data from every cell analyzed in this paper are provided as. m, and. csv files.

Before performing detailed statistical analysis, we first investigated if the data came from standard normal distributions, using two-sample Kolmogorov-Smirnov (KS) test. When we tested the mean neural activity in the response windows (intervals described below), all of our data sets were found to be distributed non-normally (non-parametric).

To show that the statistically significant cells in both experiments were not just one side of a symmetric distribution we calculated a memory index to quantify the degree to which each cell manifested memory activity, comparing postsaccadic activity in Block 1 ($R_{pre}$) with postsaccadic activity in Block 3 ($R_{mem}$):

$$\mathrm{Memory\ index} = (R_{mem} - R_{pre})/(R_{mem} + R_{pre})$$

All index distributions were not normally distributed (by KS test), and we showed that the population median was significantly different from zero by using median values for the distribution of the memory indices and a Wilcoxon signed rank test to compare each sample with a test sample identical to the measured sample with the measured mean subtracted from each value. In addition, we

calculated medcouple, a non-parametric histogram skewness measure (*Brys et al., 2004*). This tool is not affected by unsymmetrical tails and outliers, and is also not based on classical skewness represented by the third moment. Since this measure does not depend on mean or standard deviation, it gives an accurate description of a skewed distribution. Its results are bound between values: −1 (left skewed), 0 (symmetric), and 1 (right skewed).

All cells reported here had both statistically significant visual and memory responses (Wilcoxon signed rank p-value for median activity between baseline and response window <0.001) (*Whitley and Ball, 2002*). We have excluded units with memory activity if median activity in Block 1 (baseline recording block) pre- and post-saccadic were statistically different (indicating eye-position modulation). In other words, even if we could find significant memory activity well beyond what can be described by either eye position or saccadic activity effect, we are not presenting them here.

To quantify the magnitude of the memory and visual responses, we did a comparison of the peri-stimulus time histograms (PSTHs) during baseline, visual, and memory trials, for identical epochs aligned on saccade onset. All PSTHs were normalized to baseline activity (block 1) and were constructed from 2 ms spike bins, smoothed using a 10 ms sliding causal filter. Baseline calculations were made in the −426 to −226 ms (pre-saccade onset) window. To quantify the decay, perisaccadic memory and visual responses, we selected the time point of peak activity in the window from 125 ms before the saccade onset (to include any predictive remapping) to 500 ms after. We then looked at the median of raw spike count over a window extending 100 ms before and after this point to calculate response latencies, as described next.

To obtain a latency value of the memory response, we used a median threshold-crossing and sustained response method. In short, this analysis compared the median activity in the baseline and response windows. We derived a 'cut off' value by determining the spike count values which would fall above at least the 75% percentile of the baseline median. By comparing the median spike counts in the response window to this 'cut off' value, we could determine when a minimum of five bins (of 2 ms each) contained a greater spike count than the 'cut off' value. We took the first of these bins as the response latency, and verified visually by looking at the corresponding time on the PSTH. Since the baseline activity can vary between different trials in the same experiment, we first chose a window within the time from fixation to −226 ms before saccade onset. Within this wide window, we made 200 ms wide sliding windows, in 1 ms steps. By finding the median spike count within these smaller windows, we picked the window closest to the start of the response window (but still had the same median as the population of these several 200 ms wide windows). This median was chosen to look at the response window in which a threshold crossing for at least 20 ms was picked as the response latency.

We could not calculate latencies for a few neurons using this method, because the baseline (pre-saccadic) memory block response was as high as post-saccadic activity. There were also a few other units whose latencies were automatically picked by the aforementioned median method but a more precise time point could be picked by doing yet another median-based analysis. For these units, we implemented Wilcoxon rank sum comparison with the baseline median and a sliding 50 ms window, stepped 1 ms, in the response epoch. The first such 20 ms window, where there was a statistical difference was observed was picked as the latency point.

We observed that the memory response continued to manifest for almost up to 100 trials after the last probe stimulus trial, and so we analyzed the 'forgetting' or 'decay trials'. While individual cell results were noisy and there were differences between cells, we normalized each cell's decay activity to its peak memory activity and fit the population data to a first order exponential (shown in figures) and quantified it using comparisons of the first and last 10-trials in a 50-trial epoch. We fit the population data similarly.

## Acknowledgements

We thank, Moshe Shalev and Girma Asfaw for veterinary care, Yana Pavlova and Vincent Sanchez for technical assistance, John Caban and Matthew Hasday for machining, Glen Duncan for electronic and computer assistance, and Latoya Palmer, Cherise Washington, and Lisa Kennelly for facilitating everything. This research was supported, in part, by grants from the Keck, Gatsby, Kavli, Zegar, and Dana Foundations and the National Eye Institute (R24 EY-015634, R21 EY-017938, R21 EY-020631, R01 EY-017039, P30 EY-019007, and R01 EY-014978 to MG, principal investigator; SS was also

supported by training grant T32-EY-13933. MS was supported by NINDS training grant from 2T32MH015174-35, Brain and Behavior Research Foundation (NARSAD) 2013 Young Investigator Award.

## Additional information

### Funding

| Funder | Grant reference number | Author |
|---|---|---|
| National Eye Institute | R24 EY-015634 | Michael E Goldberg |
| National Institute of Neurological Disorders and Stroke | 2T32MH015174-35 | Mulugeta Semework |
| W. M. Keck Foundation | | Michael E Goldberg |
| Gatsby Charitable Foundation | | Michael E Goldberg |
| Fight for Sight | | Michael E Goldberg |
| Dana Foundation | | Michael E Goldberg |
| Kavli Foundation | | Michael E Goldberg |
| National Eye Institute | R21 EY-017938 | Michael E Goldberg |
| National Eye Institute | R21 EY-020631 | Michael E Goldberg |
| National Eye Institute | R01 EY-017039 | Michael E Goldberg |
| National Eye Institute | P30 EY-019007 | Michael E Goldberg |
| National Eye Institute | R01 EY-014978 | Michael E Goldberg |

The funders had no role in study design, data collection and interpretation, or the decision to submit the work for publication.

### Author contributions

Mulugeta Semework, Michael E Goldberg, Conceptualization, Resources, Data curation, Software, Formal analysis, Supervision, Funding acquisition, Validation, Investigation, Visualization, Methodology, Writing—original draft, Project administration, Writing—review and editing; Sara C Steenrod, Conceptualization, Data curation, Software, Formal analysis, Investigation, Visualization, Methodology, Writing—original draft, Writing—review and editing

### Author ORCIDs

Mulugeta Semework http://orcid.org/0000-0002-6070-0119
Sara C Steenrod https://orcid.org/0000-0002-7932-7385
Michael E Goldberg http://orcid.org/0000-0002-0728-2464

### Ethics

Animal experimentation: The Animal Care and Use Committees at Columbia University and the New York State Psychiatric Institute approved all of the animal protocols in this study as complying with the guidelines established in the United States Public Health Service Guide for the Care and Use of Laboratory Animals. protocol NYSPI-1225-C

### Decision letter and Author response

Decision letter https://doi.org/10.7554/eLife.30762.015
Author response https://doi.org/10.7554/eLife.30762.016

# Additional files

## Supplementary files

• Source Code 1. All Matlab Scripts for data analysis and preparation The zip file contains all Matlab scripts used in converting raw spike and analog data from NIH's REX files to Matlab, analyzing neural activities such as firing rates, decay activities etc. and creating spreadsheets and figures.
DOI: https://doi.org/10.7554/eLife.30762.011

• Source data 1. Table describing response attributes for all neurons included in the analyzed population The table provides all measured response attributes for the neurons which showed statistically significant memory activity. The first row (headers) contains the particular measure used to quantify activity such as mean, median, etc. It also includes other descriptors such as, the file name, block type, p-values for various comparisons, etc.
DOI: https://doi.org/10.7554/eLife.30762.012

• Transparent reporting form
DOI: https://doi.org/10.7554/eLife.30762.013

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
