## [Decision Letter]

Thank you for submitting your article "A spatial memory signal shows that the parietal cortex has access to a craniotopic representation of space" for consideration by *eLife*. Your article has been reviewed by three peer reviewers, one of whom is a member of our Board of Reviewing Editors and the evaluation has been overseen by Sabine Kastner as the Senior Editor. The reviewers have opted to remain anonymous.

The reviewers have discussed the reviews with one another and the Reviewing Editor has drafted this decision to help you prepare a revised submission.

Summary:

This study examined the ability of neurons in area LIP of monkeys to encode the memorized location of visual stimuli. The basic result is similar to previous findings from the senior author showing that visually responsive neurons in the frontal eye field (FEF) can respond to manipulations that relate to "environmental memory" (Umeno and Goldberg, 2001). Specifically, the neurons respond to the remembered location of a previously presented visual probe following a saccadic eye movement that aligns a given neuron's spatial receptive field to the probe location, even when the probe is now absent. However, here the authors note that the task design from the previous study made it impossible to determine if "the effect is a true spatial memory or a merely a memory of receptive field stimulation by a saccade." They therefore use new task designs, including one that dissociates saccade vectors and probe stimulation location, to distinguish these alternatives. They also focus on a different brain area (LIP vs. FEF).

The reviewers had much to praise about this study, including its clever tasks that improve upon the design of the earlier study, interesting and novel results, and a thorough discussion. However, the enthusiasm of all three reviewers for this study and its potential impact were substantially limited by several major concerns, detailed below.

Essential revisions:

1) By far the biggest concern is that the data are in many cases not presented in a manner that allows them to be evaluated effectively. Other than an initial pair of PSTH and raster plots for example responses from Task 1, the remaining results are presented primarily using scatter plots, which does not give any sense of the time course of neural responses in the context of fairly involved sensory/memory/motor sequences. Among the issues raised that should be addressed are:

a) Show time courses of responses for all three tasks, for single neurons and population averages. These plots could help to show, for example, whether there were any visually evoked responses in Task 2, Block 2; what is the time course of the responses relative to the saccade, and when the "peak activity in the baseline and memory blocks of trials" occurred in Task 3.

b) Account for some very high spike rates – some were as high as 400+ sp/sec. Could this reflect multi-unit activity?

c) A better justification for using the medc metric, as opposed to more straightforward tests comparing distributions.

2) Much of the data from this paper comes from a single monkey, leading to two issues. First, tasks 1 and 2 have data from two other monkeys that appropriately and effectively confirm the more extensive results from the first monkey. However, it is unusual to treat data from monkey B and C as if from a single monkey. At the very least, it would be useful to use different symbols for the two monkeys (e.g., red triangle & red square) in the scatterplots, to demonstrate that the results are similar for the two. Second, results from Task 3 are only from one monkey, which does not meet the acknowledged standard in this field. It therefore is important to make it clear that this is an unconfirmed, preliminary finding that is not critical to the overall message of the study. Alternatively, confirmatory data from a second monkey is needed.

3) The writing and figures should be revised carefully for clarity. For example, in the current manuscript it is difficult to understand the details of the tasks, in particular key timing issues. The way the figures are organized does not help. For example, it is not clear whether probe stimulus is ON when the monkey makes the saccades (1 or 2) during PROBE ON trials of block 3 and 2 of task 1 and 2, respectively. It would be easy to add under each task figure the time course of each block/trials. Other figures have errors and/or other issues that should be fixed. For example, Figure 5, block 2 probe ON, the position of the fixation point and target are moved to the right compared to other figures. Figure 7 block 1, an arrow says the fovea is at the center of the screen.

4) The interpretation of the results could also use further clarification, particularly with respect to a "craniotopic reference frame." In particular, it is not clear that the results allow for this claim. This study did not involve manipulations of the orientation of the head, and the information about the position of the probe could be encoded in any other reference frame (hand, body). As the authors certainly know, it could also be "an absolute, world-based frame of reference" (Colby & Duhamel, 1996). Moreover, the relationship to lesion findings should also be clarified. For example, what is basis for the claim that the study by Bisiach and Luzzatti shows that "retrieving that memory goes through a retinotopic process that requires the parietal cortex"? Finally, it would be useful to clarify what these and other data say about the specific role of LIP in generating spatial representations, as distinguished from other parts of the oculomotor network including the FEF.

[Editors' note: further revisions were requested prior to acceptance, as described below.]

Thank you for resubmitting your work entitled "A spatial memory signal shows that the parietal cortex has access to a craniotopic representation of space" for further consideration at *eLife*. Your revised article has been favorably evaluated by Sabine Kastner (Senior editor) and a Reviewing editor.

The manuscript has been improved but there are some remaining issues that need to be addressed before acceptance, as outlined below:

1) The most major point is that it is not clear that the results from Task 2 justify the claim that the environmental memory signal "does not require visual stimulation." Inspection of Figure 8 suggests the presence of a visual response in the "Form memory" condition, at least for Monkeys B and C. Neural responses for the Block 2, stimulus vs no-stimulus condition should be compared directly if this claim is to be supported.

2) The additional transparency about data from Monkeys B and C in the figures is appreciated. However, I would suggest not using the statement: 'Since the results from Monkeys B and C were comparable and we had only a small number of cells in each monkey, we combined them into "Monkey B".' Instead, perhaps say something like 'Because the results from Monkeys B and C were comparable and we had only a small number of cells in each monkey, several analyses combine these two data sets.'

Then for the text and figures (e.g., Figure 4), use "Monkeys B and C" instead of "Monkey B"

3) Figure 5 and Figure 9 are pretty hard to parse, and it is not clear what the normalization to Block 1 adds. Perhaps consider: (1) removing those points if they are not necessary, or (2) using a supplemental figure if they are, along with a stronger justification for including them. Also, is a cubic polynomial justified for fitting the open symbols in 5d?

4) How should the lack of decay in Monkeys B and C for Task 2 be interpreted?

[Editors' note: further revisions were requested prior to acceptance, as described below.]

Thank you for resubmitting your work entitled "A spatial memory signal shows that the parietal cortex has access to a craniotopic representation of space" for further consideration at *eLife*. Your revised article has been favorably evaluated by Sabine Kastner (Senior editor) and a Reviewing editor.

The manuscript has been improved but there still is one remaining issue that needs to be addressed before acceptance, as outlined below:

The added information describing the neural responses to Task 2 is welcome but awfully cryptic, and I'm not sure it clearly indicates that the environmental memory signal "does not require visual stimulation." Specifically:

1) Identifying a statistically significant difference between Block 2 with stimulus and Block 4 does not seem either surprising or interesting: in Block 4, the stimulus was placed in the RF and so by definition should produce a bigger response.

2) The comparison of memory recall versus memory formation is key, as noted previously, but here treated in a cursory manner: p-values for population analyses do not give any intuition for size and/or timing of spike-rate differences for each neuron (and p-values of 0.02 and 0.03 should not be considered as evidence for a "highly statistical difference"), nor does this analysis rule out that there were visual responses in the probe-on trials.

3) For the population baseline-vs-memory block analysis, a p-value of 0.05 is not considered significant, even with outliers, and so this should not be considered the "same" result.

Can the authors provide more clear and definitive evidence that the environmental memory signal "does not require visual stimulation"?

Please note that we strongly expect you to provide conclusive and convincing evidence to resolve these concerns satisfactorily, since we are unlikely to grant another round of revisions after this one.

---

## [Author Response]

Essential revisions:

1) By far the biggest concern is that the data are in many cases not presented in a manner that allows them to be evaluated effectively. Other than an initial pair of PSTH and raster plots for example responses from Task 1, the remaining results are presented primarily using scatter plots, which does not give any sense of the time course of neural responses in the context of fairly involved sensory/memory/motor sequences. Among the issues raised that should be addressed are:

We now show time courses of responses for two tasks, for single neurons and population averages. The third task been entirely removed to address the concern that the data came from a single monkey.

a) Show time courses of responses for all three tasks, for single neurons and population averages. These plots could help to show, for example, whether there were any visually evoked responses in Task 2, Block 2; what is the time course of the responses relative to the saccade, and when the "peak activity in the baseline and memory blocks of trials" occurred in Task 3.

Corrected, as described above. There are visually-evoked responses in Task 2 only in the last block and the time course of the responses around the saccade resemble those from Task 1.

b) Account for some very high spike rates – some were as high as 400+ sp/sec. Could this reflect multi-unit activity?

These are not multi-unit activities but actual peak spiking rates calculated from PSTHs in the response window. These rates are high because they are maximums and are calculated from PSTHs which are constructed from narrow bins (2 ms) using a smoothing window (10 ms). In the current version, we have normalized the spiking activity to that of Block 1 (baseline) pre-saccadic activity and there are no longer such extreme peak firing rates.

c) A better justification for using the medc metric, as opposed to more straightforward tests comparing distributions.

Medc is used for non-Gaussian distributions. For the sake of simplicity, and to provide extra confirmation we now have added a straightforward median calculation. We left medc in for the benefit of readers who wish to have extra validation of the result.

2) Much of the data from this paper comes from a single monkey, leading to two issues. First, tasks 1 and 2 have data from two other monkeys that appropriately and effectively confirm the more extensive results from the first monkey. However, it is unusual to treat data from monkey B and C as if from a single monkey. At the very least, it would be useful to use different symbols for the two monkeys (e.g., red triangle & red square) in the scatterplots, to demonstrate that the results are similar for the two. Second, results from Task 3 are only from one monkey, which does not meet the acknowledged standard in this field. It therefore is important to make it clear that this is an unconfirmed, preliminary finding that is not critical to the overall message of the study. Alternatively, confirmatory data from a second monkey is needed.

These are also valid points. Therefore: (1) we have clearly identified monkey B differently from monkey C by adding ‘*’ on top of the red triangles in the scatter plots. 2) Task 3, being from only one monkey, has been completely removed. It will be used in future publications when a second monkey is recorded from.

3) The writing and figures should be revised carefully for clarity. For example, in the current manuscript it is difficult to understand the details of the tasks, in particular key timing issues. The way the figures are organized does not help. For example, it is not clear whether probe stimulus is ON when the monkey makes the saccades (1 or 2) during PROBE ON trials of block 3 and 2 of task 1 and 2, respectively. It would be easy to add under each task figure the time course of each block/trials. Other figures have errors and/or other issues that should be fixed. For example, Figure 5, block 2 probe ON, the position of the fixation point and target are moved to the right compared to other figures. Figure 7 block 1, an arrow says the fovea is at the center of the screen.

All of these issues have been addressed by replacing the task figures with completely new ones. The new versions include cartoons illustrating task timing, associated eye-positions, and text defining the role of each block of trials, such as “baseline”, “form memory”, “recall”, etc.

4) The interpretation of the results could also use further clarification, particularly with respect to a "craniotopic reference frame." In particular, it is not clear that the results allow for this claim. This study did not involve manipulations of the orientation of the head, and the information about the position of the probe could be encoded in any other reference frame (hand, body). As the authors certainly know, it could also be "an absolute, world-based frame of reference" (Colby & Duhamel, 1996).

The reviewers are correct that from the evidence in this paper we cannot assert that LIP has access to a craniotopic representation. However, our data clearly argue that LIP has access to a more general representation of visual space than a retinotopic one. To eliminate the unwarranted specificity of “craniotopic” we have substituted “supraretinal” for “craniotopic” throughout the paper.

Moreover, the relationship to lesion findings should also be clarified. For example, what is basis for the claim that the study by Bisiach and Luzzatti shows that "retrieving that memory goes through a retinotopic process that requires the parietal cortex"?

We now write,

“Our results can explain previous clinical observations regarding the role of parietal cortex in spatial memory. Patients with bilateral parietal lesions cannot point to an object in their room with their eyes closed, although they can easily do it with their eyes open (Farah et al., 1985). This suggests that these patients do not have access to a remembered spatial representation of their environment. More importantly, patients with right parietal lesions report their spatial memory in a retinotopic frame. Bisiach and Luzzatti (1978) asked two Milanese patients with right parietal damage to describe their memory of the Piazza del Duomo in Milan. When they described it as if they were standing with their back to the cathedral they remembered only the landmarks to the right of the cathedral. When they described it as if they were facing the cathedral they only remembered the landmarks on the other side of the square, which were now to the right of their imagined vantage point. This experiment demonstrates two key ideas; 1. The brain stores long-term memory of environments in supraretinal coordinates in an area unaffected by a parietal lesion, since the patients could, under certain circumstances, remember the entire Piazza del Duomo. 2. Retrieving that memory requires the parietal cortex, but the right parietal patients could only report (and therefore image) objects on the right side of their imagined vantage point. They had a mnemonic left hemianopia for a remembered environment. If their memory were reportable in a supraretinal frame they should have been able to report the entire scene. Because they could not do so, but could only report objects in the non-neglected field, the report of their environmental memory must be processed by the retinotopic map of the parietal cortex. Our results are consistent with this result. The memory of an environment can be stored a supraretinal frame, but must be reported by neurons with retinotopic receptive fields. The neurons that we have reported here fit that description.”

Finally, it would be useful to clarify what these and other data say about the specific role of LIP in generating spatial representations, as distinguished from other parts of the oculomotor network including the FEF.

We now write,

“It is important to emphasize that LIP is part of a spatial network that includes the FEF and the superior colliculus. Although the neuropsychological studies suggest that parietal damage affects spatial memory, we do not know if damage to other parts of the network has a similar effect.”

[Editors' note: further revisions were requested prior to acceptance, as described below.]

1) The most major point is that it is not clear that the results from Task 2 justify the claim that the environmental memory signal "does not require visual stimulation." Inspection of Figure 8 suggests the presence of a visual response in the "Form memory" condition, at least for Monkeys B and C. Neural responses for the Block 2, stimulus vs no-stimulus condition should be compared directly if this claim is to be supported.

Another great observation to which we provide the following answers:

It is true that Figure 8 shows a perisaccadic burst in the memory trials of block 3. However, it also shows a perisaccadic burst in block 1, when no stimulus had ever appeared in the receptive field of the neuron and the saccade did not bring the stimulus into the receptive field. LIP neurons often have a small perisaccadic burst when monkeys make any saccade (Wang and Goldberg, in preparation), and the perisaccadic burst.

We now write (Results section):

"We studied 54 neurons (30 in Monkey A, 14 in Monkey C) in the No-RF task. We used the same criteria as for the neurons studied in the basic task, with one major difference: some LIP neurons have a brief perisaccadic response to eye movements in all directions, including those away from their receptive fields. Rather than excluding neurons with perisaccadic responses, we included neurons with perisaccadic responses that did not differ between blocks 1 (saccades that will, in block 3, bring the spatial location of the vanished stimulus into the receptive field) and block 3 (saccades made away from the receptive field)."And:

"To investigate whether, as a population, the groups showed statistically significant memory responses, we utilized two approaches. First, we compared population PSTHs for the different blocks. [...] We observed the environmental memory response (Figure 9) despite the facts that 1) the probe stimulus never appeared in the neuron's receptive field, and 2) the saccade used to establish the memory response (saccade 1)."

2) The additional transparency about data from Monkeys B and C in the figures is appreciated. However, I would suggest not using the statement: 'Since the results from Monkeys B and C were comparable and we had only a small number of cells in each monkey, we combined them into "Monkey B".' Instead, perhaps say something like 'Because the results from Monkeys B and C were comparable and we had only a small number of cells in each monkey, several analyses combine these two data sets.'Then for the text and figures (e.g., Figure 4), use "Monkeys B and C" instead of "Monkey B"

Corrected and figure legends and text references.

Now, the data for second task (Task 2) comes from only Monkey C. This helped in creating a population that is consistent and representative.

3) Figure 5 and Figure 9 are pretty hard to parse, and it is not clear what the normalization to Block 1 adds. Perhaps consider: 1) removing those points if they are not necessary, or 2) using a supplemental figure if they are, along with a stronger justification for including them. Also, is a cubic polynomial justified for fitting the open symbols in 5d?

In the spirit of full disclosure about the history of these panels, the main idea behind adding block 1 normalization and showing both came mainly from questions from one specific researcher in our floor. We do accept that this approach complicated the message, and thus have simplified them by:

1) Removing the extra plots and axis i.e. we now show only the decay vs last few memory forming trials.

2) Using only a first order exponential fit. Cubic polynomial was used to help readers follow the modulation and also get a better fit to the data (r-square). It is worth noting here that our main criteria for measuring decay is statistical comparison between the first and last 10-trials in a given epoch. As such, using the wrong fit can cause non-optimal line fits as is evidenced by a negative r-square for Monkey A Task 2 fit. We can take out the r-square value references if the editor feels it will confuse readers.

3) We have expanded our decay epoch to 50 trials (instead of 40) to show better (comparably sharper) decay at least for one of the monkeys(groups).

4) How should the lack of decay in Monkeys B and C for Task 2 be interpreted?

We found that most of our units “forget” in about 40 trials, meaning, the lack of decay for the specific data set and task is only for the 40-trial epoch, not for the whole “forgetting” block. As mentioned above, we now show 50-trial epoch confirming decay, albeit slow. There are units that can keep their memory even for up to 100 trials. To explain this further showing up to 100 trials is not practical as:

1) Not all neurons keep their memory that long.

2) Not all recordings have 100 such trial.

These two points mean that we will have enough units for 100 trials to average over.

We now say (Subsection “Task 2: No – RF Task.”):

“For this task, Monkey A showed no decay within the 50-trial epoch. This is common for some units in the population which can keep their memory even for up to 100 trials. Since not all recordings have a large number of ‘forgetting’ trials, it is not possible to have enough units for 100 trials to average over. The negative R2 demonstrates the slowness of the decay and that a first-order exponential is not the best fit (it caused non-optimal line fit). This fit is used in the figures for consistency and visual clarity.”

On a previous and somewhat unclear statement we made about latencies: we now show latencies for all units using different approaches. We now say (Subsection “Task 1 – Basic Memory Task”):

“However, because the memory responses for some neurons increased gradually, we were unable to establish memory latencies for all neurons using our latency method (for these neurons, we used other approaches, as described in the Materials and methods section).”

This refers to what we say (Materials and methods section):

[Editors' note: further revisions were requested prior to acceptance, as described below.]

The added information describing the neural responses to Task 2 is welcome but awfully cryptic, and I'm not sure it clearly indicates that the environmental memory signal "does not require visual stimulation." Specifically:1) Identifying a statistically significant difference between Block 2 with stimulus and Block 4 does not seem either surprising or interesting: in Block 4, the stimulus was placed in the RF and so by definition should produce a bigger response.

We respectfully disagree – this result just shows that the memory response is weaker (and has a longer latency) than the pure visual response. Although not a main point of the paper it does say something about the memory phenomenon and is worth including in the paper. Furthermore, the larger visual response in Block 4 than in Block 2 shows that whatever decay we saw in Block 3 was not due to our losing the cell over this rather long experiment. Finally, we do not use the Block 2-Block 4 comparison to establish that the memory response was not a visual response. We now write (Subsection “Task 2: No-RE task.”).

“As in the basic task, the larger visual response in Block 4 was larger than the memory response in Block 2 (Monkey A, p < 0.0001, Monkey C p = 0.03 Wilcoxon signed rank). This shows that whatever decay we saw in Block 3 was not due to our losing the cell over this rather long experiment. The memory response had a longer latency than the neuron’s visual response (Figure 9) (Wilcoxon signed rank p = 0.008 for Monkey A, and 0.006 for Monkey C; mean visual/memory latency: 39/101 ms for Monkey A, 25/106 ms for Monkey C).”

2) The comparison of memory recall versus memory formation is key, as noted previously, but here treated in a cursory manner: p-values for population analyses do not give any intuition for size and/or timing of spike-rate differences for each neuron (and p-values of 0.02 and 0.03 should not be considered as evidence for a "highly statistical difference").

The scatter plots (Figure 5 and Figure 9) give a precise measure of the differences in peak spike rates for each cell, and the latency of the cell’s response. The population histograms give an estimate of timing across the population.

Nor does this analysis rule out that there were visual responses in the probe-on trials.

The data that show we do not have a visual response in the no-RF experiment is the comparison between Block 1 and Block 2. There is no visual stimulus in Block 1, nor has there ever been a visual stimulus in the RF at that point in the experiment. There is a visual stimulus outside the RF in Block 2 trials with the stimulus. This stimulus has no effect on the cell’s activity, because there is no difference between the small postsaccadic response in Block 1, and the small postaccadic response in Block 2 when the monkey makes a saccade in the different direction that does not bring the stimulus into the receptive field. Therefore. the response in block 2 cannot be a visual response. We now write, (Subsection “Task 2: No-RE task.”)

“Therefore, rather than excluding neurons with postsaccadic responses, we included neurons with small postsaccadic responses that did not differ between blocks 1 (saccades that will, in block 2, bring the spatial location of the vanished stimulus into the receptive field, but in which no stimulus ever appeared on the screen) and block 2 (different saccades that brought the receptive field away from the stimulus whose memory is being established, Wilcoxon signed rank: Monkey A p-value = 0.4; Monkey C p-value = 0.3). Although the stimulus never appeared in the neuron’s receptive field, the cells responded when the saccade brought the spatial location of the vanished stimulus into the receptive field (Figure 7). The population exhibited a robust memory response (Figure 8, poststimulus histogram for all cells we studied). The population histogram shows, for one monkey, a small postsaccadic burst in the stimulus appearing trials of block 2. One could argue, by looking at the postsaccadic response in the trials where the monkey made the saccade that brought the visual stimulus to a location outside of our estimate of the cell’s receptive field, that this postsaccadic response was a visual response.”

3) For the population baseline-vs-memory block analysis, a p-value of 0.05 is not considered significant, even with outliers, and so this should not be considered the "same" result.

We realized that we had been using the wrong nonparametric test for most of our analysis. We are now using the Wilcoxon Signed Rank test, the non-parametric equivalent of the paired t-test, for the population analyses. Using this test, we have much lower p values for the population studies. The only p < 0.05 value we used was to establish memory in single neurons, by showing differences in median activity in the postsaccadic epoch for Block 1 trials trials, before the memory had been established, and the stimulus-absent trials of Block 2, which had the memory response even though the task was identical in in the two blocks, the monkey making a saccade that brings the receptive field into a specific location where there is no stimulus. For a single cell to show environmental memory, the difference in the response had to be significant, p < 0.05 by Wilcoxon Rank Sum, which is identical to the Mann Whitney U test, the nonparametric t test. The use of p < 0.05 for single cells is pretty standard across the literature, for example Joshi et al., (2016).

Can the authors provide more clear and definitive evidence that the environmental memory signal "does not require visual stimulation"?

Please see our reply to point 2 above.